# The Impact of the Pandemic on Mental Health in Ethnically Diverse Mothers: Findings from the Born in Bradford, Tower Hamlets and Newham COVID-19 Research Programmes

**DOI:** 10.3390/ijerph192114316

**Published:** 2022-11-02

**Authors:** Claire McIvor, Yassaman Vafai, Brian Kelly, Sarah E. O’Toole, Michelle Heys, Ellena Badrick, Halima Iqbal, Kate E. Pickett, Claire Cameron, Josie Dickerson

**Affiliations:** 1Faculty of Health Studies, The University of Bradford, Bradford BD7 1DP, UK; 2Department of Health Sciences, University of York, York YO10 5DD, UK; 3UCL Institute of Education, University College London, London WC1E 6BT, UK; 4East London NHS Foundation Trust, London E1 8DE, UK; 5UCL Population, Policy and Practice Research and Teaching Department, University College London Great Ormond Street Institute of Child Health, London WC1N 1EH, UK

**Keywords:** COVID-19, mental health, ethnicity, ethnic minorities, health inequalities, deprivation

## Abstract

Restrictions implemented by the UK Government during the COVID-19 pandemic have served to worsen mental health outcomes, particularly amongst younger adults, women, those living with chronic health conditions, and parents of young children. Studies looking at the impact for ethnic minorities have reported inconsistent findings. This paper describes the mental health experiences of mothers from a large and highly ethnically diverse population during the pandemic, using secondary analysis of existing data from three COVID-19 research studies completed in Bradford and London (Tower Hamlets and Newham). A total of 2807 mothers participated in this study with 44% White British, 23% Asian/Asian British Pakistani, 8% Other White and 7% Asian/Asian British Bangladeshi backgrounds. We found that 28% of mothers experienced clinically important depressive symptoms and 21% anxiety symptoms during the pandemic. In unadjusted analyses, mothers from White Other, and Asian/Asian British Bangladeshi backgrounds had higher odds of experiencing symptoms, whilst mothers from Asian/Asian British Indian backgrounds were the least likely to experience symptoms. Once loneliness, social support and financial insecurity were controlled for, there were no statistically significant differences in depression and anxiety by ethnicity. Mental health problems experienced during the pandemic may have longer term consequences for public health. Policy and decision makers must have an understanding of the high risk of financial insecurity, loneliness and a lack of social support on mother’s mental health, and also recognise that some ethnic groups are far more likely to experience these issues and are, therefore, more vulnerable to poor mental health as a consequence.

## 1. Introduction

The UK Government lockdown measures imposed to control the spread of the COVID-19 pandemic have been shown to have unintended negative consequences on the health of the UK population [1]. These measures have affected some groups more than others, with those from ethnic minorities or socioeconomically deprived backgrounds experiencing the greatest negative impact on their physical health, mental health, financial insecurity and food insecurity [2,3,4].

The closure of workplaces and schools during the early phases of the pandemic in the UK caused increased financial insecurity, particularly in those working in low paid/self-employed roles and those having to juggle childcare and work [2,5]. The link between financial insecurity and mental ill health is well established, and increased prevalence of mental ill health during the pandemic has been reported for those who are socially and economically disadvantaged [5,6]. Minority ethnic groups were found to be at increased risk of being hospitalised and of death from COVID-19, with those from Black African, Black Caribbean, Pakistani, Bangladeshi, and Indian backgrounds identified as most at risk of the disease when compared to White British or White Irish backgrounds [7]. This increased fear of becoming ill or dying of the disease and served to worsen mental health outcomes amongst those from ethnic minority backgrounds [8]. Women from ethnic minority backgrounds have been identified in some studies as the most vulnerable to mental-ill health during the pandemic, with those from Black, Asian, or Hispanic backgrounds reporting worse mental health outcomes than individuals who identified as White [9].

In addition, the pandemic and associated lockdown measures served to further accentuate gender disparities, especially for employed mothers or single parent mothers due to the disproportionate responsibility placed on women for domestic duties and childcare [10].

Poor mental health during the COVID-19 pandemic has also been reported as having a higher prevalence among those living alone, younger adults (≤40 years), women, those living with chronic physical or mental health conditions, and parents of young children [11].

We have conducted three studies on the impact of the COVID-19 restrictions on mothers from a range of ethnic backgrounds, with a range of socio-economic status, and found varied prevalence between ethnic groups and differing associations between ethnicity and mental health during the pandemic [2,12,13].

In a longitudinal study in Bradford comparing levels of depression and anxiety from before, to during the pandemic in mothers from mainly Pakistani heritage (48%) or White British backgrounds (34%), we found increases in the number of mothers reporting clinically significant symptoms [14] with higher rates in mothers from White British compared to Pakistani heritage. An increase in poor mental health was associated with loneliness, financial, food, and housing insecurity, a lack of physical activity, and a poor partner relationship. Once these variables were controlled for, there were no clear differences in the increases in poor mental health by ethnic group. There was however variation in the magnitude of the associations by ethnicity. For example, compared to White British mothers, those from a Pakistani heritage had greater odds of an increase in depression and anxiety if they were lonely or had an average/poor relationship and had much-reduced odds of an increase in depression if they lived in a large household. In contrast, mothers of White British ethnicity had greater odds of an increase in depression if they were financially insecure and/or physically inactive compared with Pakistani heritage mothers [14].

In two cross sectional studies in London, based in Tower Hamlets (36% Bangladeshi, 34% White British) and Newham (12.9% Bangladeshi, 6.3% Black African, 16.9% Other White) higher incidences of clinically important depression and anxiety were reported by mothers from Black (42%) and Bangladeshi (21%) backgrounds compared to other ethnic groups (White British/Irish: 18%, Asian Other: 15%). Similar to the Bradford mothers, poor mental health was associated with financial, food and housing insecurity, loneliness, a lack of social support, an average/poor partner relationship and a lack of physical activity.

To explore the potential associations between ethnicity and mental health in more detail we combined and analysed cross-sectional data from our three uniquely ethnically diverse areas in England, all with high rates of socio-economic deprivation: Bradford, Tower Hamlets, and Newham. The combined data across these three areas allows for more nuanced exploration of the association between a range of ethnic groups and mental health for mothers. The objectives of this study are to:Explore what factors are associated with poor mental health during the COVID-19 pandemic across a range of ethnic groups in three areas in England.Explore in depth, the association between financial insecurity and mental health.Explore the association between loneliness and mental health.Identify how any ethnic differences in mother’s mental health are modified by financial insecurity, loneliness, and social support.

## 2. Materials and Methods

### 2.1. Setting

Two of the studies are nested within the ActEarly Research programme: Born in Bradford and Families in Tower Hamlets, and the third is the Families in Newham study. The ActEarly research programme is designed to work with local communities and authorities to understand how to help families to live healthier and more active lives [15]. The study teams collaborated to collect similar measures (All three programmes were linked to ActEarly (2019–2024), a UKPRP funded network to leverage research and evaluation to improve health and life chances of children). The three study areas are characterised by ethnic diversity and high variance in levels of financial insecurity. Bradford has a young, ethnically diverse population with the largest proportion of people of Pakistani heritage (20.3%) in England [16]. In Tower Hamlets, more than two thirds (69%) of the borough’s population are from minority ethnic groups and it is the 16th most ethnically diverse local authority in England [17], whilst in Newham, just under half (48%) of residents were born outside of the UK, with the proportion of residents identifying as Bangladeshi (12.4%), Black African (11.1%), Indian (14.8%), and Pakistani heritage (9.8%) greater than the London average [18].

### 2.2. Study Design and Participants

#### 2.2.1. Bradford

The Born in Bradford participants were drawn from the Born in Bradford COVID-19 study within which existing cohort participants were invited to complete longitudinal surveys to understand the impact of the pandemic [2,19].

In the first survey (March 2020–June 2020), a total of 2043 mothers who had children aged 0 to 11 participated.

#### 2.2.2. Tower Hamlets

The Tower Hamlets participants were drawn from Families in Tower Hamlets, a study of the impacts of COVID-19 on families with young children or expecting a baby that took place between 2020–2022. Parents were recruited through general and targeted borough communications channels and asked to complete an online (Qualtrics) survey. Data reported here are from the first wave of the survey (July–November 2020) with a total of 732 mothers. Full details of this study have been published previously [12].

#### 2.2.3. Newham

The Families in Newham study ran in parallel to Families in Tower Hamlets and involved the Qualtrics survey used in the Families in Tower Hamlets study. Recruitment was via borough public health personnel. Data reported here are from the survey that ran from August–December 2020 and included 1252 mothers.

### 2.3. Data Collection

Participants were recruited to all studies using a combination of emails, text, and phone, and in their main language wherever possible. Surveys were completed by participants online, on the telephone, or using postal surveys.

### 2.4. Measures

#### 2.4.1. Outcomes

Mental health information was self-reported using the PHQ-8 for depressive symptoms [20] and the GAD-7 for anxiety symptoms [21]. PHQ-8 and GAD-7 are validated instruments and widely used to measure the severity of symptoms in depression and anxiety in the general population and ethnic minority populations in the U.K.

#### 2.4.2. Exposures

##### Ethnicity

The 2011 UK Census ethnic categories were used to recategorize ethnicity into 10 groups including White British, White Irish, other White, Black African, Black Caribbean, Asian Indian, Asian Pakistani, Asian Bangladeshi, other Asian, and other ethnic groups [22].

##### Financial Insecurity

Information on financial insecurity was self-reported using the question ‘How well would you say you are managing financially right now?’ with response options including ‘living comfortably’, ‘doing alright’, ‘just about getting by’, ‘finding it quite difficult’, ‘finding it very difficult’, ‘don’t know’, and ‘prefer not to answer’.

##### Loneliness

Loneliness was self-reported using the question ‘How often have you felt lonely during the past week?’ with answer options including: ‘none, or almost none of the time’, ‘some of the time’, ‘most of the time’, ‘all or almost all of the time’, ‘don’t know’, and ‘prefer not to answer’.

##### Other Household Characteristics

Information on all other household factors including relationship status and quality, pregnancy status, living in a damp or mouldy house, worrying about current employment status, food insecurity, and presence of social support at the time of survey (including the number of people you can count on) was also self-reported and details on response options are presented in Table 1.

##### Location of Residency

For the purpose of this study, for the final analysis Tower Hamlets and Newham were combined into one location comparator referred to as ‘London’.

### 2.5. Data Analysis

To combine the three datasets, we created a 3-level cohort variable (Bradford, Tower Hamlets or Newham) and the location of residency variable. The questions asked in all three surveys were linked together and categorised as follows:

Depression & Anxiety measures: We used total scores and standard categories of PHQ-8 (0–4 no depressive symptoms, 5–9 mild depressive symptoms, 10–14 moderate depressive symptoms, 15–19 moderately severe depressive symptoms, and 20–24 severe depressive symptoms) and GAD-7 (0–4 no anxiety symptoms, 5–9 mild anxiety symptoms, 10–14 moderate anxiety symptoms, and 15–21 severe anxiety symptoms). These categories were collapsed into no clinically significant symptoms (none or mild); and clinically significant symptoms (moderate, moderately severe, and severe) for the final analysis.

Ethnicity variables were collapsed into seven categories including: White British, White Other (all other White categories), Asian/Asian British Indian, Asian/Asian British Pakistani, Asian/Asian British Bangladeshi, Black/Black British Caribbean/African, and Other ethnic groups.

Financial insecurity was reclassified into a binary variable as secure (living comfortably/doing alright) and insecure (just getting by/finding it quite difficult/finding it very difficult). Loneliness was also reclassified as never lonely (none/almost none of the time), sometimes lonely (some of the time) or always lonely (Most/all or almost all of the time) for the final analysis.

Participants with missing data on any of the variables were excluded from all analyses.

Descriptive statistics were calculated to compare maternal and household characteristics for different levels of depressive and anxiety symptoms using proportions and the corresponding 95% confidence intervals (CI). To further explore the potential contributing factors to mothers’ mental health, we estimated the odds of developing clinically important depressive symptoms for ethnicity, location of residency, financial insecurity, loneliness, and social support using univariate logistic regression models. We then examined the individual effect of location of residency, financial insecurity, loneliness, and social support on the association between ethnicity and clinically important depressive symptoms using four individual multivariate regression models.

Furthermore, to explore the combined effect of ethnicity and financial insecurity on the development of clinically important depressive symptoms during the COVID-19 pandemic, we conducted a regression model to estimate the predictive probability [23] of experiencing clinically important depressive symptoms among different ethnic groups adjusting for location of residency and financial insecurity with an interaction term between ethnicity and financial insecurity. Finally, since social support has been shown to be an important factor in improving mental health, especially among certain ethnic minorities [24], we added the social support variable and an interaction term between ethnicity and social support. We repeated all the steps, replacing financial insecurity with loneliness to explore the combined effect of loneliness on the relationship between ethnicity and experiencing clinically important depressive symptoms. We fitted a regression model with all the aforementioned variables and interaction terms included. We repeated all the analysis steps for anxiety symptoms. The odds ratios, the predictive probabilities (percentages) of clinically important depressive/anxiety symptoms and the corresponding 95% CIs are presented. All analyses were conducted in STATA (version 16.)

## 3. Results

### 3.1. Sample Characteristics

The combined dataset included 4024 mothers. Of these, 1217 were excluded from the analysis, leaving a total of 2807 mothers with 52% (*n* = 1466) from Bradford, 16% (*n* = 445) from Tower Hamlets, and 32% (*n* = 896) from Newham. The mean depressive symptoms score was 6.4 (SD = 5.3) ranging from 0–24 and 28% (*n* = 777) of mothers experienced clinically important depressive symptoms. The mean anxiety symptoms score was 5.7 (SD = 5.0) ranging from 0–21 and 21% (*n* = 586) of mothers experienced clinically important anxiety symptoms (Table 1).

Of the 2807 participants, 44% (*n* = 1237) were White British, 23% (*n* = 659) were of Asian/Asian British Pakistani heritage, 8% (*n* = 219) were of other White background and 7% (*n* = 200) were of Asian/Asian British Bangladeshi heritage (Figure 1).

**Table 1 ijerph-19-14316-t001:** Sample characteristics overall and by study site.

Characteristics	Overall		Bradford		Tower Hamlets		Newham	
	*n* = 2807	%	*n* = 1466	%	*n* = 445	%	*n* = 896	%
Depressive symptoms								
None	1236	44%	876	60%	154	35%	206	23%
Mild	794	28%	344	23%	136	31%	314	35%
Moderate	561	20%	148	10%	90	20%	323	36%
Moderately severe	163	6%	73	5%	51	11%	39	4%
Severe	53	2%	25	2%	14	3%	14	2%
Anxiety symptoms								
None	1360	48%	930	63%	182	41%	248	28%
Mild	861	31%	335	23%	138	31%	388	43%
Moderate	421	15%	120	8%	82	18%	219	24%
Severe	165	6%	81	6%	43	10%	41	5%
Ethnicity								
White								
British	1237	44%	609	42%	176	40%	452	50%
Irish	121	4%	<5	---	<5	---	116	13%
Any other White	219	8%	39	3%	58	13%	122	14%
Black/Black British/Mixed Caribbean/African								
Caribbean	46	2%	21	1%	8	2%	17	2%
African	35	1%	16	1%	12	3%	7	1%
Asian/Asian British/Mixed								
Indian	94	3%	51	3%	11	2%	32	4%
Pakistani	659	23%	623	43%	8	2%	28	3%
Bangladeshi	200	7%	37	3%	129	29%	34	4%
Any other Asian	81	3%	32	2%	18	4%	31	4%
Any other ethnic group/Mixed	115	4%	36	2%	22	5%	57	6%
Relationship status								
Single	328	12%	188	13%	67	15%	73	8%
Married/civil partnership	2194	78%	1117	76%	329	74%	748	83%
Not married but in a relationship	285	10%	161	11%	49	11%	75	8%
Pregnancy status								
No	2515	90%	1415	97%	369	83%	731	82%
Yes	292	10%	51	3%	76	17%	165	18%
Home condition—mould or damp house								
No	2114	75%	1090	74%	322	72%	702	78%
Yes	693	25%	376	26%	123	28%	194	22%
Worrying about job security								
Strongly disagree	383	14%	272	19%	33	7%	78	9%
Disagree	691	25%	392	27%	69	16%	230	26%
Neither agree or disagree	659	24%	296	20%	133	30%	230	26%
Agree	735	26%	338	23%	150	34%	247	28%
Strongly agree	339	12%	168	11%	60	13%	111	12%
Food insecurity—Food didn’t last								
Never true	2061	73%	1243	85%	271	61%	547	61%
Sometimes true	512	18%	174	12%	133	30%	205	23%
Often true	234	8%	49	3%	41	9%	144	16%
Food insecurity—Couldn’t afford balanced meals								
Never true	2072	74%	1238	84%	265	60%	569	64%
Sometimes true	474	17%	156	11%	135	30%	183	20%
Often true	261	9%	72	5%	45	10%	144	16%
Food insecurity—Have been hungry?								
No	2467	88%	1413	96%	376	84%	678	76%
Yes	340	12%	53	4%	69	16%	218	24%
Financial insecurity—how are you getting on								
Living comfortably	597	21%	342	23%	82	18%	173	19%
Doing alright	1172	42%	666	45%	158	36%	348	39%
Just about getting by	728	26%	341	23%	126	28%	261	29%
Finding it quite difficult	212	8%	85	6%	47	11%	80	9%
Finding it very difficult	98	3%	32	2%	32	7%	34	4%
Quality of relationship with partner								
NA- Single	328	12%	188	13%	67	15%	73	8%
Excellent	1002	36%	643	44%	133	30%	226	25%
Good	1051	37%	508	35%	166	37%	377	42%
Average	348	12%	103	7%	52	12%	193	22%
Poor	50	2%	14	1%	17	4%	19	2%
Very poor	28	1%	10	1%	10	2%	8	1%
Social support—No of people you can count on								
0–2: Low	683	24%	273	19%	131	29%	279	31%
3–6: Medium	1397	50%	651	44%	235	53%	511	57%
7 and more: High	727	26%	542	37%	79	18%	106	12%
Social support—No of people you can count on living locally								
0–2: Low	1386	49%	573	39%	253	57%	560	63%
3–6: Medium	1069	38%	608	41%	165	37%	296	33%
7 and more: High	352	13%	285	19%	27	6%	40	4%
Loneliness								
None/almost none of the time	1313	47%	899	61%	163	37%	251	28%
Some of the time	995	35%	450	31%	198	44%	347	39%
Most of the time	384	14%	84	6%	50	11%	250	28%
All/almost all of the time	115	4%	33	2%	34	8%	48	5%

Characteristics of mothers by categories of depressive symptoms are presented in Table 2. Overall, during the COVID-19 pandemic, mothers of Asian/Asian British Pakistani, Bangladeshi and Black/Black British Caribbean/African backgrounds, single mothers, pregnant women, mothers living in damp houses, mothers worried about job security, mothers experiencing food insecurity and financial insecurity, mothers with low levels of social support, and those experiencing loneliness were more likely to experience clinically important depression.

### 3.2. Factors Associated with Clinically Important Depressive Symptoms

To explore the association between ethnicity, location of residency, financial insecurity, loneliness, social support, and depressive symptoms during the COVID-19 pandemic, we conducted a series of logistic regression models among participants. In the univariate models, being from a White other, Black/Black British Caribbean/African, and Asian/Asian British Bangladeshi background, residing in London, feeling lonely most of the time, having low or medium levels of social support and being financially insecure were significantly associated with experiencing clinically important depressive symptoms during the pandemic (Table 3). The odds of experiencing clinically important depressive symptoms remained significant only for mothers of White other ethnicity compared to White British mothers after adjusting for location of residency (OR = 1.88, 95% CI = 1.45, 2.43), level of loneliness (OR = 1.81, 95% CI = 1.37, 2.39), social support (OR = 2.23, 95% CI = 1.73, 2.88), and financial insecurity (OR = 2.36, 95% CI = 1.83, 3.04) (Table 3). In the unadjusted models the odds of experiencing clinically important depressive symptoms was significantly lower for Asian/Asian British Indian mothers (OR = 0.43, 95% CI = 0.23, 0.78) compared to White British mothers and remained unchanged after adjusting for the location of residency (OR = 0.43, 95% CI = 0.34, 0.80), social support (OR= 0.43, 95% CI = 0.23, 0.79) and financial insecurity (OR= 0.43, 95% CI =0.24, 0.79). Similarly, the odds of experiencing clinically important depressive symptoms were significantly lower for Asian/Asian British Pakistani mothers than White British mothers after adjusting for social support (OR= 0.59, 95% CI = 0.47, 0.75) and financial insecurity (OR= 0.52, 95% CI = 0.41, 0.66).

### 3.3. Ethnic Differences in the Risk of Clinically Important Depressive Symptoms and Role of Financial Insecurity

To examine whether financial insecurity during the COVID-19 pandemic modified the risk of experiencing depressive symptoms for different ethnic groups, we estimated the predictive probability of experiencing clinically important depressive symptoms and the corresponding 95% CIs for each ethnic group (Figure 2 and Appendix A). In the unadjusted model, mothers from White other (49.7%, 95% CI: 44.4–55.0%) and Asian/Asian British Bangladeshi (34.5%, 95% CI: 27.9–41.1%) backgrounds had the highest chance of experiencing clinically important depressive symptoms compared to other ethnic groups, while Asian/Asian British Indian (13.8%, 95% CI: 6.9%, 20.8%) and Asian/Asian British Pakistani (17.8%, 95% CI: 14.8%, 20.7%) mothers had the lowest chance. (Figure 2, Panel a; Appendix A). In the adjusted model for financial insecurity only, the predictive probability of experiencing clinically important depressive symptoms remained significant for mothers from White other backgrounds only (47.7%, 95% CI: 42.5%, 52.9%) (Figure 2, Panel b; Appendix A). Mothers from the White other background (39%, 95% CI: 33.9%, 44%) continued to have a statistically significant higher predictive probability of experiencing clinically important depressive symptoms compared to White British, Asian/Asian British Indian, Pakistani, Bangladeshi, and women from other ethnic backgrounds, in the model adjusted for location of residency, financial insecurity, and the interaction between ethnicity and financial insecurity (Figure 2, Panel c; Appendix A). The addition of social support, and an interaction term between ethnicity and social support to the regression model did not change the ethnic differences seen in the previous models and mothers from White other background continued to have a significantly higher predictive probability of clinically important depressive symptoms compared to mothers of White British, Asian/British Asian Indian, Pakistani, Bangladeshi, and other ethnic backgrounds (Figure 2, Panel d; Appendix A).

### 3.4. Ethnic Differences in the Risk of Clinically Important Depressive Symptoms and the Role of Loneliness

We repeated the analyses to explore the potential modifying role of loneliness in the relationship between ethnicity and depressive symptoms (Figure 3; Appendix A). In the regression model adjusted for loneliness alone, there were no statistically significant differences in the predictive probability of clinically important depressive symptoms among ethnic groups (Figure 3, Panel a; Appendix A). Similarly, the addition of location of residency and an interaction term between ethnicity and loneliness (Figure 3, Panel b; Appendix A) produced no significant differences between ethnic groups. Finally, the addition of social support and an interaction term between ethnicity and social support did not change the findings (Figure 3, Panel c; Appendix A).

Finally, we ran a regression model adjusting for location of residency, financial insecurity, loneliness, and social support with interaction terms between ethnicity and financial insecurity, loneliness, and social support also included. In this model, there were no differences for the risk of experiencing clinically important depressive symptoms between ethnic groups (Figure 4).

### 3.5. Clinically Important Anxiety Symptoms

We repeated all analyses to estimate the predictive probability of clinically important anxiety and found very similar results (Appendix A).

## 4. Discussion

This study examined the association between ethnicity and mental health during the COVID-19 pandemic in a large and highly ethnically diverse population, whilst controlling for potential confounding socio-economic circumstances. We found that 28% of mothers experienced clinically important depressive symptoms, and 21% reported clinically important symptoms of anxiety. These findings reflect those of other studies which have reported similar instances of clinically important symptoms of depression and anxiety amongst mothers during the COVID-19 pandemic [9,11,25]. Rates of clinically important symptoms of depression and anxiety were higher in those who reported feeling lonely, having low or medium levels of social support and/or being financially insecure.

In unadjusted analyses, mothers from a White other, and Asian/Asian British Bangladeshi backgrounds had higher odds of experiencing clinically important symptoms during the pandemic, whilst mothers from Asian/Asian British Indian backgrounds were the least likely to experience symptoms. However, once loneliness, social support and financial insecurity were controlled for, no statistically significant differences were found between the ethnic groups. This finding suggests that the important factors associated with depression and anxiety during the pandemic are loneliness, a lack of social support and financial insecurity. Any effects reported by ethnicity are most likely a consequence of the fact that mothers in some ethnic groups are far more likely to be lonely, lack social support and/or be financially insecure. Ethnic differences in household structure and support networks may explain in part why these differences exist, as differing cultural practices and local networks mean that some ethnic groups had less support and financial resilience during the pandemic than others [24,26].

An interesting and unanticipated finding was the increased likelihood of depression and anxiety symptoms in the London compared to the Bradford participants. This may again be due to increased isolation, less availability of social support and higher living costs in London compared to the relatively close-knit Bradford communities where the cost of living is also lower. However, there were also small differences in the timing of the surveys, the population who participated, and there may also be other unidentified confounders which were not identified in this analysis. Nevertheless, this finding warrants further investigation, in particular in terms of any long-lasting increases in poor mental health for mothers residing in London, particularly in light of the cost-of-living crisis unfolding in the post-pandemic era.

A strength of this study is that the women included were from a wide range of ethnically diverse backgrounds. Many research studies group ethnic populations under the umbrella terms of ‘BAME’ or ‘South Asian’ due to a lack of diversity in their samples; however, a major strength of this study is the heterogeneity of participants. This allowed us to gain nuanced insight into the mental health experiences of those from often under-researched and under-represented backgrounds. We also collected data from three different locations within England strengthening the generalisability of our findings.

Whilst a strength of our study is the diversity of the sample, there are relatively small numbers included in some ethnic groups. This therefore may have limited the power of the study to detect statistically significant differences in the probability of experiencing clinically important symptoms of depression or anxiety in some ethnic groups. Furthermore, the cross-sectional design of this study does not allow us to determine any causation of our results. We did not include age of the mothers in the sample as an exposure, as this variable was not available in the Newham sample, however all mothers had a child aged 0–11 years meaning there will not be huge variation by age. We do however recognise this as a limitation of the study.

Research studies describing the associations between ethnicity and mental health during the COVID-19 pandemic are limited. Future research should focus on understanding the longer-term impacts of the pandemic on mental health, particularly in light of ongoing changes in social and economic circumstances in the UK. By conducting inclusive research which attempts to understand the experiences of those from minority ethnic groups, we have demonstrated here the ability to better understand and support the needs of those who have struggled during the COVID-19 pandemic and into the future.

## 5. Conclusions

The COVID-19 pandemic has served to exacerbate existing inequalities, having a greater impact on those already vulnerable. The cost-of-living crisis has placed further challenges on families, with financial insecurity rising and limitations on family’s opportunities to socialise.

Our results show that financial insecurity, loneliness, levels of social support, and location of residency were all associated with clinically important depression and anxiety during the pandemic. Once these key variables were controlled for, there were no differences in symptoms by ethnicity.

These findings speak to two main policy areas: (i) poverty and inequality of income; and (ii) support available through community and neighbourhoods. Policy and decision makers must have an understanding of these factors when considering methods to support vulnerable families as the Government begins to implement plans on ‘Levelling up’ during the post pandemic recovery [27]. They should also be aware that some ethnic groups are far more likely to experience these issues, that this may have a negative impact on their mental wellbeing, and that support targeting these key groups would be of great benefit.

## Figures and Tables

**Figure 1 ijerph-19-14316-f001:**
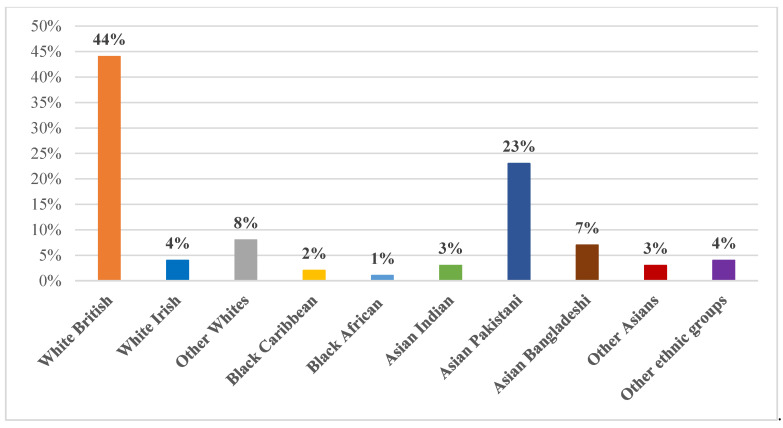
The overall ethnic distribution of the sample (*n* = 2807).

**Figure 2 ijerph-19-14316-f002:**
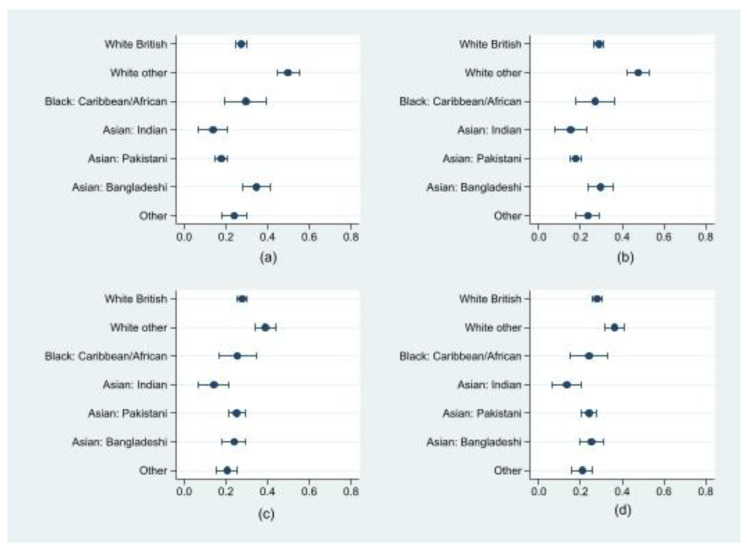
Predictive probability of clinically important depressive symptoms by ethnicity (*n* = 2807); (**a**) unadjusted model for the association between ethnicity and depressive symptoms; (**b**) regression model for the association between ethnicity and depressive symptoms adjusted for financial insecurity (secure (ref), insecure); (**c**) regression model for the association between ethnicity and depressive symptoms adjusted for location of residency (Bradford (ref), London) and financial insecurity (secure (ref), insecure) and interaction between ethnicity and financial insecurity; (**d**) regression model for the association between ethnicity and depressive symptoms adjusted for location of residency (Bradford (ref), London), financial insecurity (secure (ref), insecure), social support (high (ref), medium, low), and interactions between ethnicity and financial insecurity, and ethnicity and social support.

**Figure 3 ijerph-19-14316-f003:**
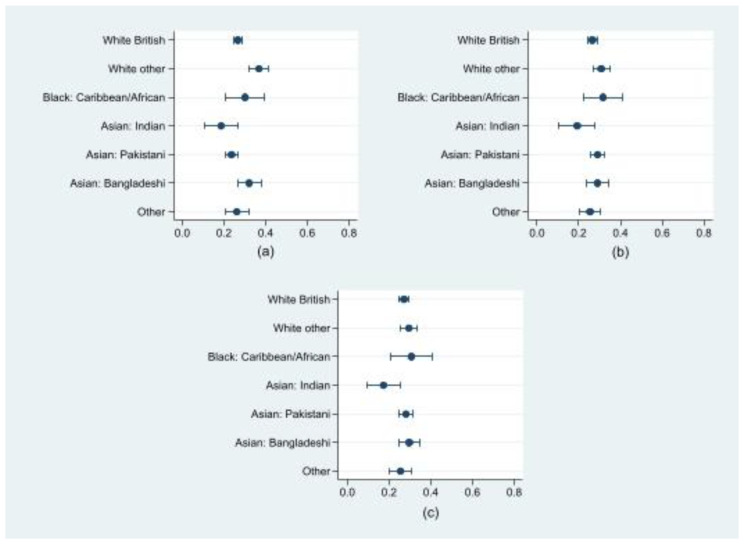
Predictive probability of clinically important depressive symptoms by ethnicity (*n* = 2807): (**a**) regression model for the association between ethnicity and depressive symptoms adjusted for loneliness (none of the time (ref), some of the time, most of the time); (**b**) regression model for the association between ethnicity and depressive symptoms adjusted for location of residency (Bradford (ref), London), loneliness (none of the time (ref), some of the time, most of the time) and interaction between ethnicity and loneliness (none of the time (ref), some of the time, most of the time); (**c**) and regression model for the association between ethnicity and depressive symptoms adjusted for location of residency (Bradford (ref), London), loneliness, and social support (high (ref), medium, low), and interaction between ethnicity and loneliness (none of the time (ref), some of the time, most of the time) and ethnicity and social support (high (ref), medium, low).

**Figure 4 ijerph-19-14316-f004:**
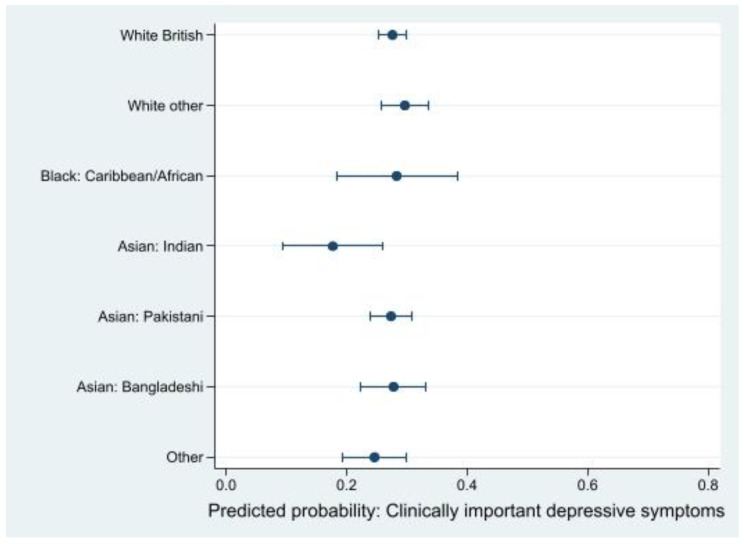
Predictive probability of clinically important depressive symptoms by ethnicity (*n* = 2807). Regression model for the association between ethnicity and depressive symptoms was adjusted for location of residency (Bradford (ref), London) with interaction terms between ethnicity and financial insecurity (secure (ref), insecure), ethnicity and loneliness (none of the time (ref), some of the time, most of the time), ethnicity and social support (high (ref), medium, low levels).

**Table 2 ijerph-19-14316-t002:** Differences in maternal and home characteristics by levels of depressive symptoms (*n* = 2807).

Characteristics	Depressive Symptoms										
	None		Mild		Moderate		Moderately Severe		Severe		Total
	*n*	% (95% CI)	*n*	% (95% CI)	*n*	% (95% CI)	*n*	% (95% CI)	*n*	% (95% CI)	*n*
Overall	1236	44% (42–46%)	794	28% (27–30%)	561	20% (19–22%)	163	6% (5–7%)	53	2% (1–2%)	2807
Ethnicity											
White											
British	462	37% (35–40%)	437	35% (33–38%)	253	20% (18–23%)	68	6% (4–7%)	17	1% (1–2%)	1237
Irish	13	11% (6–18%)	21	17% (12–25%)	84	69% (61–77%)	<5	---	<5	---	121
Any other White	82	37% (31–44%)	55	25% (20–31%)	67	31% (25–37%)	12	5% (3–9%)	5	1% (1–4%)	219
Black/British Black											
Caribbean	23	50% (36–64%)	8	17% (9–31%)	7	15% (7–29%)	5	11% (5–24%)	<5	---	46
African	17	49% (33–65%)	9	26% (14–42%)	<5	---	<5	---	<5	---	35
Asian/British Asian											
Indian	57	61% (50–70%)	24	26% (18–35%)	8	9% (4–16%)	5	5% (2–12%)	0	---	94
Pakistani	415	63% (59–67%)	127	19% (16–22%)	69	10% (8–13%)	35	5% (4–7%)	13	2% (1–3%)	659
Bangladeshi	71	36% (29–42%)	60	30% (24–37%)	41	21% (15–27%)	22	11% (7–16%)	6	3% (1–7%)	200
Any other Asian	43	53% (42–64%)	21	26% (18–37%)	11	14% (8–23%)	6	5% (2–12%)	<5	---	81
Any other ethnic group/Mixed	53	46% (37–55%)	32	28% (20–37%)	17	15% (9–23%)	6	5% (2–11%)	7	6% (3–12%)	115
Relationship status											
Single	130	40% (34–45%)	89	27% (23–32%)	49	15% (11–19%)	43	13% (10–17%)	17	5% (3–8%)	328
Married/civil partnership	980	45% (43–47%)	618	28% (26–30%)	462	21% (19–23%)	108	5% (4–6%)	26	1% (1–2%)	2194
Not married but in a relationship	126	44% (39–50%)	87	31% (25–36%)	50	18% (14–22%)	12	4% (2–7%)	10	4% (2–6%)	285
Pregnancy status											
No	1147	46% (44–48%)	733	29% (27–31%)	435	17% (16–19%)	150	6% (5–7%)	50	2% (2–3%)	2515
Yes	89	30% (25–36%)	61	21% (17–26%)	126	43% (38–49%)	13	4% (3–8%)	<5	---	292
Home condition—mould or damp house											
No	976	46% (44–48%)	629	30% (28–32%)	384	18% (17–20%)	97	5% (4–6%)	28	1% (1–2%)	2114
Yes	260	38% (34–41%)	165	24% (21–27%)	177	26% (22–29%)	66	10% (8–12%)	25	4% (2–5%)	693
Worrying about job security											
Strongly disagree	219	57% (52–62%)	80	21% (17–25%)	53	14% (11–18%)	23	6% (4–9%)	8	2% (1–4%)	383
Disagree	339	49% (45–53%)	182	26% (23–30%)	139	20% (17–23%)	25	4% (2–5%)	6	1% (0.4–2%)	691
Neither agree or disagree	271	41% (37–45%)	175	27% (23–30%)	155	24% (20–27%)	39	6% (4–8%)	19	3% (2–4%)	659
Agree	297	40% (37–44%)	237	32% (29–36%)	146	20% (17–23%)	48	7% (5–9%)	7	1% (0.4–2%)	735
Strongly agree	110	32% (28–38%)	120	35% (30–41%)	68	20% (16–25%)	28	8% (6–12%)	13	4% (2–6%)	339
Food insecurity—Food didn’t last											
Never true	1074	52% (50–54%)	616	30% (28–32%)	274	13% (12–15%)	78	4% (3–5%)	19	1% (1–2%)	2061
Sometimes true	132	26% (22–30%)	139	27% (23–31%)	172	34% (30–38%)	48	9% (7–12%)	21	4% (3–6%)	512
Often true	30	13% (9–18%)	39	17% (12–22%)	115	49% (43–56%)	37	16% (12–21%)	13	6% (3–9%)	234
Food insecurity—Couldn’t afford balanced meals											
Never true	1065	51% (49–54%)	620	30% (28–32%)	286	14% (12–15%)	81	4% (3–5%)	20	1% (1–2%)	2072
Sometimes true	125	26% (23–31%)	124	26% (22–30%)	158	33% (29–38%)	48	10% (8–13%)	19	4% (3–6%)	474
Often true	46	18% (13–23%)	50	19% (15–24%)	117	45% (39–51%)	34	13% (9–18%)	14	5% (3–9%)	261
Food insecurity—Have been hungry?											
No	1208	49% (47–51%)	734	30% (28–32%)	386	16% (14–17%)	105	4% (4–5%)	34	1% (1–2%)	2467
Yes	28	8% (6–12%)	60	18% (14–22%)	175	51% (46–57%)	58	17% (13–21%)	19	6% (4–9%)	340
Financial insecurity—How are you getting on?											
Living comfortably	368	62% (58–65%)	151	25% (22–29%)	56	9% (7–12%)	21	4% (2–5%)	<5	---	597
Doing alright	566	48% (45–51%)	322	27% (25–30%)	224	19% (17–21%)	47	4% (3–5%)	13	1% (1–2%)	1172
Just about getting by	236	32% (29–36%)	247	34% (31–37%)	184	25% (22–29%)	45	6% (5–8%)	16	2% (1–4%)	728
Finding it quite difficult	48	23% (18–29%)	56	26% (21–33%)	69	33% (27–39%)	27	13% (9–18%)	12	6% (3–10%)	212
Finding it very difficult	18	18% (12–27%)	18	18% (12–27%)	28	29% (21–38%)	23	23% (16–33%)	11	11% (6–19%)	98
Relationship quality with partner											
NA-Single	130	40% (34–45%)	89	27% (23–32%)	49	15% (11–19%)	43	13% (10–17%)	17	5% (3–8%)	328
Excellent	602	60% (57–63%)	237	24% (21–26%)	125	12% (11–15%)	24	2% (2–4%)	14	1% (1–2%)	1002
Good	433	41% (38–44%)	336	32% (29–35%)	224	21% (19–24%)	47	4% (3–6%)	11	1% (1–2%)	1051
Average	63	18% (14–23%)	119	34% (29–39%)	137	39% (34–45%)	24	7% (5–10%)	5	1% (1–3%)	348
Poor	5	10% (4–22%)	9	18% (10–31%)	18	36% (24–50%)	16	32% (21–46%)	<5	---	50
Very poor	<5	---	<5	14% (5–32%)	8	29% (15–48%)	9	32% (18–51%)	<5	---	28
Social support—No. of people you can count on											
0–2: Low	196	29% (25–32%)	206	30% (27–34%)	183	27% (24–30%)	76	11% (9–14%)	22	3% (2–5%)	683
3–6: Medium	582	42% (39–44%)	414	30% (27–32%)	310	22% (20–24%)	71	5% (4–6%)	20	1% (1–2%)	1397
7 and more: High	458	63% (59–66%)	174	24% (21–27%)	68	9% (7–12%)	16	2% (1–4%)	11	2% (1–3%)	727
Social support—No. of people you can count on living locally											
0–2: Low	490	35% (33–38%)	470	34% (31–36%)	282	20% (18–23%)	110	8% (7–9%)	34	2% (2–3%)	1386
3–6: Medium	523	49% (46–52%)	255	24% (21–27%)	236	22% (20–25%)	43	4% (3–5%)	12	2% (1–2%)	1069
7 and more: High	223	63% (58–68%)	69	20% (16–24%)	43	12% (9–16%)	10	3% (2–5%)	7	2% (1–4%)	352
Loneliness											
None/almost none of the time	891	68% (65–70%)	288	22% (20–24%)	106	8% (7–10%)	22	2% (1–3%)	6	0.5% (0.2–1%)	1313
Some of the time	314	32% (29–35%)	371	37% (34–40%)	2401	24% (22–27%)	60	6% (5–8%)	10	1% (1–2%)	995
Most of the time	26	7% (5–10%)	113	29% (25–34%)	77	46% (41–51%)	47	12% (9–16%)	21	5% (4–8%)	384
All/almost all of the time	5	4% (2–10%)	22	19% (13–27%)	38	33% (25–42%)	34	30% (22–39%)	16	14% (9–22%)	115

**Table 3 ijerph-19-14316-t003:** Associations between ethnicity and clinically important depressive symptoms (*n* = 2807).

	Univariate	Multivariate			
	Unadjusted	Adjusted for Location of Residency Only	Adjusted for Loneliness Only	Adjusted for Social Support Only	Adjusted for Financial Insecurity Only
	OR (95% CI)	OR (95% CI)	OR (95% CI)	OR (95% CI)	OR (95% CI)
Ethnicity					
White other	2.63	1.88	1.81	2.23	2.36
	(2.05–3.36)	(1.45–2.43)	(1.37–2.39)	(1.73–2.88)	(1.83–3.04)
Black/British Black: Caribbean/African	1.12	1.09	1.24	1	0.91
	(0.68–1.83)	(0.65–1.80)	(0.71–2.18)	(0.59–1.64)	(0.55–1.51)
Asian/British Asian: Indian	0.43	0.43	0.57	0.43	0.43
	(0.23–0.78)	(0.24–0.80)	(0.30–1.09)	(0.23–0.79)	(0.24–0.79)
Asian/British Asian: Pakistani	0.57	0.97	0.82	0.59	0.52
	(0.45–0.73)	(0.74–1.26)	(0.64–1.07)	(0.47–0.75)	(0.41–0.66)
Asian/British Asian: Bangladeshi	1.4	1.05	1.39	1.35	1.04
	(1.02–1.92)	(0.76–1.45)	(0.98–1.98)	(0.97–1.87)	(0.75–1.44)
Other	0.84	0.72	0.97	0.78	0.75
	(0.59–1.19)	(0.50–1.03)	(0.66–1.43)	(0.55–1.12)	(0.52–1.08)
White British	1	1	1	1	1
Location of residency					
London	3.25	2.87			
	(2.73–3.87)	(2.32–3.54)			
Bradford	1	1			
Loneliness					
Some of the time	3.98		3.67		
	(3.18–4.98)		(2.93–4.61)		
All of the time	17.65		15.31		
	(13.64–22.85)		(11.76–19.93)		
None of the time	1		1		
Social Support					
Medium (3–6)	2.68			2.43	
	(2.10–3.42)			(1.90–3.12)	
Low (0–2)	4.65			4.08	
	(3.57–6.06)			(3.11–5.34)	
High (+7)	1			1	
Financial insecurity					
Insecure	2.59				2.57
	(2.19–3.07)				(2.15–3.06)
Secure	1				1

## Data Availability

ESRC funded survey data is available from UKDS; https://beta.ukdataservice.ac.uk/datacatalogue/studies/study?id=855477. Newham data is not covered by archived arrangements and is available on application.

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
