# Peer review of "The Impact of the Pandemic on Mental Health in Ethnically Diverse Mothers: Findings from the Born in Bradford, Tower Hamlets and Newham COVID-19 Research Programmes"

_ijerph, 2022, doi:10.3390/ijerph192114316_

Round 1
Reviewer 1 Report
- Clear purpose of the paper however please include (in the abstract) the reason of such study group (mothers) chosen for the research.
- Well described the process of recruitment of group as well as the process of data collection and analysis.
- Beware that Table 2 and 3 are invisible and therefore hard to read and review.
- I am not the specialist on depression (symptoms as well as reasons), however please provide a clear link (theoretical background) between depression symptoms and financial insecurity (as I understand depression leads to many other problems and financial ones are only one of several possible problems). You could also clearly provide the information on how your result differ from other studies conducted worldwide and how important the role of financial aspects is. What is more, you can add the information on why the financil stability is important for human being (based on other theoretical findings).
- I do appreciate the statistical methods as well as other tolls used for analysis.
- You have provided the further ways of research or suggestions for other researchers willing to use your idea for other, extended studies, which may lead to other studies in the future.
- Please consider to present the most important findings as well as differences between particular groups in a table or bullet points (it may be easier for the reader to notice the most crucial findings and conclusion of that paper).
Reviewer 2 Report
1. Although this research has many sources of funding, as well as the participation of many research institutions and scholars, the content of the article is relatively rough. It is suggested that the quality of the article, at least the layout, should be enhanced. The authors of the articles are mostly scholars from English-speaking research institutions, and it is recommended to demonstrate better English writing skills. Much of the content (eg 2.4) has a very rough layout. The content of some tables is not even complete. The support of such a wealth of research resources is not commensurate with the quality of this article.
2. At least the format of the authors' contact information should be consistent.
3. Please rethink the meaning of keywords. For example, keywords such as "Born in Bradford; Tower Hamlets; Newham" are inappropriate.
4. Introduction: Please reduce self-citations. And please introduce more relevant research theories and results, instead of just using many pages to introduce the authors' past research results. In addition, it is not appropriate to use "we" as the subject of the sentence. After all, not all of the six authors of this article are the authors of the cited literature.
5. In terms of sampling, the so-called minority groups account for 56% of the total sample. Is this reasonable? Can it accurately reflect the population?
6. The resolution of the pictures is really poor.
7. Only a small amount of relevant theory is provided. The format of References is inconsistent. In addition, please reduce the citation of technical reports and Internet materials. Inappropriate citations should also be improved. For example, as for the 19th reference, not only the format and content are wrong, is it appropriate to choose a one-page research report in 2014? Why not directly cite the research results of Spitzer et al. in 2006 to introduce the GAD-7 questionnaire?
Reviewer 3 Report
Dear Author/s,
The covid19 pandemic and its consequences affected globally every aspect of health, including mental health and psychological behavior. Many factors are included in this unwanted life-threatening pandemic. Many scientific studies published on disease biology, behavior, economic well-being, affordability for medication, and panic status will remain for several years in the minds of people around the globe.
1. This study is a good attempt to understand the impact of psychological variables on mental health during the covid-19. Furthermore, this study has included the symptoms related to depression and anxiety. It has also included other variables which can impact mental health such as loneliness and job security.
2. In line 126, the author mentioned “Full details are available elsewhere”, this sentence can be rephrased.
3. Nothing is mentioned about the routine activities of the women during the lockdown.
4. No information is mentioned about all the mentioned factors in the pre-lockdown pandemic era.
5. Authors have used ‘black’ and ‘white’ very often, although it doesn't seem racial but if they can explore other possibilities e.g. ‘Black Caribbean’, why not only ‘Caribbean’?
6. Line 91-97; The objectives of the study formatting of writing in the paragraph are changed? The authors should include that in similar fonts/phrases
7. Figure 2/3: The figure needs clarity. Add statistics to the figure to understand group difference
8. Figure legends and text in the figure has different size of fonts
9. Table 1 and Table 2 should be in a different orientation to cover all the columns visible within the page format.
